# Wide Next-Generation Sequencing Characterization of Young Adults Non-Small-Cell Lung Cancer Patients

**DOI:** 10.3390/cancers14102352

**Published:** 2022-05-10

**Authors:** Paola Ulivi, Milena Urbini, Elisabetta Petracci, Matteo Canale, Alessandra Dubini, Daniela Bartolini, Daniele Calistri, Paola Cravero, Eugenio Fonzi, Giovanni Martinelli, Ilaria Priano, Kalliopi Andrikou, Giuseppe Bronte, Lucio Crinò, Angelo Delmonte

**Affiliations:** 1Biosciences Laboratory, IRCCS Istituto Romagnolo per lo Studio dei Tumori (IRST) “Dino Amadori”, 47014 Meldola, Italy; paola.ulivi@irst.emr.it (P.U.); milena.urbini@irst.emr.it (M.U.); daniele.calistri@irst.emr.it (D.C.); 2Unit of Biostatistics and Clinical Trials, IRCCS Istituto Romagnolo per lo Studio dei Tumori (IRST) “Dino Amadori”, 47014 Meldola, Italy; elisabetta.petracci@irst.emr.it (E.P.); eugenio.fonzi@irst.emr.it (E.F.); 3Pathology Unit, “Morgagni-Pierantoni” Hospital, 47121 Forli, Italy; alessandra.dubini@auslromagna.it; 4Pathology Unit, “Maurizio Bufalini” Hospital, 47521 Cesena, Italy; daniela.bartolini@auslromagna.it; 5Department of Medical Oncology, IRCCS Istituto Romagnolo per lo Studio dei Tumori (IRST) “Dino Amadori”, 47014 Meldola, Italy; paola.cravero@irst.emr.it (P.C.); giovanni.martinelli@irst.emr.it (G.M.); ilaria.priano@irst.emr.it (I.P.); kalliopi.andrikou@irst.emr.it (K.A.); giuseppe.bronte@irst.emr.it (G.B.); lucio.crino@irst.emr.it (L.C.); angelo.delmonte@irst.emr.it (A.D.)

**Keywords:** NGS, NSCLC, targeted alterations, multitarget

## Abstract

**Simple Summary:**

Molecular characterization of advanced non-small-cell lung cancer (NSCLC) is mandatory before any treatment decision making. Next-generation sequencing (NGS) approaches represent the best strategy in this context. In our study, we analyzed a case series of young (under 65 years old) NSCLC patients with a wide NGS gene panel assay. The most frequent altered genes were TP53 (64.55%), followed by KRAS (44.1%), STK11 (26.9%), CDKN2A (21.5%), CDKN2B (14.0%), EGFR (16.1%), and RB1 (10.8%). Tumor mutational burden (TMB) was also evaluated considering different cut-offs, and we found a significant association between TMB and STK11 and KRAS mutations. Conversely, EGFR and EML4-ALK alterations were more frequently found in tumors with low TMB. We compared results obtained from this approach with those obtained from a single or few genes approach, observing perfect concordance of the results.

**Abstract:**

Molecular characterization of advanced non-small-cell lung cancer (NSCLC) is mandatory before any treatment decision making. Next-generation sequencing (NGS) approaches represent the best strategy in this context. The turnaround time for NGS methodologies and the related costs are becoming more and more adaptable for their use in clinical practice. In our study, we analyzed a case series of young (under 65 years old) NSCLC patients with a wide NGS gene panel assay. The most frequent altered genes were TP53 (64.55%), followed by KRAS (44.1%), STK11 (26.9%), CDKN2A (21.5%), CDKN2B (14.0%), EGFR (16.1%), and RB1 (10.8%). Tumor mutational burden (TMB) was also evaluated. Considering the cut-off of 10 mut/Mb, 62 (68.9%) patients showed a TMB < 10 mut/Mb, whereas 28 (31.1%) showed a TMB ≥ 10 mut/Mb. STK11 and KRAS mutations were significantly associated with a higher TMB (*p* = 0.019 and *p* = 0.004, respectively). Conversely, EGFR and EML4-ALK alterations were more frequently found in tumors with low TMB (*p* = 0.019 and *p* < 0.001, respectively). We compared results obtained from this approach with those obtained from a single or few genes approach, observing perfect concordance of the results.

## 1. Introduction

Lung cancer is the leading cause of cancer-related mortality, with an estimation of 2.22 million new cases and 1.55 million deaths per year worldwide [1]. Lung cancer incidence is influenced by environmental and lifestyle factors, such as smoking habits, diet and alcohol consumption, and air pollution, which are able to act on the mutational status of susceptibility genes (i.e., *TP53*) [2,3]. Although it is considered a disease that affects elderly people, with a median age of 70 years, about 1–10% of patients at diagnosis present median age of 40 years, with a poor outcome [4,5]. Young patients with non-small-cell lung cancer (NSCLC) have specific clinicopathological features. They are more frequently female, non-smokers, with adenocarcinoma (ADC) histology, and present at diagnosis a lymph node positivity and an invasive disease [4,5,6]. From a molecular point of view, NSCLC in young patients seems to be characterized by a different genetic profile, with the observation of several targetable alterations, representing important options in treatment [7].

Molecular characterization of NSCLC patients is nowadays mandatory prior to any treatment decision making. “Precision medicine” refers to the use of drugs more likely to confer benefit to a subgroup of patients whose cancer carries certain molecular alterations. Driver alterations have been identified in about 70% of adenocarcinoma (ADC) patients, involving principally the EGFR, KRAS, HER2, BRAF, RET, MET, ALK, ROS1, and NTRK genes [8], and specific targeted drugs are already used in clinical practice, whereas others are under investigation in clinical trials [9]. The growing number of approved and emerging pharmacological options, together with the continuous evolution of molecular biology technologies, have resulted in a rapid change of therapeutic algorithms for NSCLC patients. This makes it difficult to standardize the guidelines for biomarker testing, and molecular testing rates remain suboptimal [10]. The European Society for Medical Oncology’s (ESMO) NSCLC guidelines recommend systematic testing of EGFR and BRAF mutations; analysis of ALK, ROS1, and NTRK rearrangements; and determination of PD-L1 expression. The guidelines also suggest the routine use of next-generation sequencing (NGS) methodologies [11], indicating that large multigene panels could be preferred instead of small panels. However, until some years ago, the majority of molecular diagnostics laboratories performed single-gene testing, using different methodologies such as MassArray, pyrosequencing, real-time PCR, and digital PCR [12]. This approach is becoming more and more inappropriate in a reality where even more gene alterations have to be evaluated in clinical practice [10]. Moreover, the entry of immunotherapy in the lung cancer treatment scenario makes it necessary to evaluate the other alterations which could have a role in the selection of patients more likely to benefit from treatment. The amount of gene alterations, evaluated as tumor mutation burden (TMB), has an emerging role (although not definitely demonstrated) in selecting those patients more responsive to immunotherapy [13]. Although contradicting conclusions have been made regarding the role of TMB as a marker of patient selection for immunotherapy, a recent review and meta-analysis showed that it represents a good prognostic marker in this setting. However, specific methodological aspects in its determination and cut-off selection have to be clarified and standardized to help its potential use in clinical practice [14]. Moreover, specific mutations in single genes are also under investigation as potential predictive biomarkers for immunotherapy [15]. These aspects reinforce the necessity to perform a wide tumor genomic characterization at diagnosis, allowing the clinician to have all the necessary information to decide the best treatment strategy.

In the present study, we aim to provide a wide molecular characterization performed by Foundation One CDX assay in NSCLC young adult patients as well as a comparison with standard methodological approaches used in clinical practice. The potentiality of a wide molecular characterization is discussed in the light of the new evidence regarding the recent interesting results about novel emerging targeted agents, as well as the emergence of specific gene mutations associated with drug resistance.

## 2. Materials and Methods

### 2.1. Case Series

A retrospective case series of 368 NSCLC patients treated at IRCCS IRST between 2004 and 2019, aged ≤ 65 years, and with appropriate histologic material was considered in this study. Inclusion criteria were cytological or histological diagnosis of NSCLC, age ≤ 65 years at diagnosis, previous diagnostic molecular characterization, and available histologic material. In particular, we selected cases for which histological formalin-fixed paraffin-embedded samples were available, with the possibility to cut at least 10 slides of 5 microns, with a tumor area percentage of at least 50%, selected by a pathologist on a hematoxylin–eosin-stained section. Clinical data were retrieved from a medical chart review. The study was approved by the Romagna Ethics Committee (CEROM), No. IRSTB095, Prot. 8437/2018, I.5/274. It was also conducted in accordance with the Declaration of Helsinki 1964 and later versions, and all patients signed the informed consent.

### 2.2. Sequencing Analysis

A next-generation sequencing-based assay (Foundation One CDx) was performed. The assay can detect substitutions, insertion, and deletion alterations (indels), and copy number alterations (CNAs) in 324 genes and selected gene rearrangements, as well as genomic signatures including microsatellite instability (MSI) and tumor mutational burden (TMB) using DNA isolated from formalin-fixed paraffin-embedded (FFPE) tumor tissue specimens.

Mutations were classified accordingly to Foundation One annotation on the predicted somatic status and functional impact. Somatic and pathogenic mutations were indicated as “known” or “likely”. Variants of unknown significance were excluded from further analysis. Among the 93 patients, 68 also had a previous characterization by standard methodologies as part of the routine molecular diagnostics evaluation performed by the Molecular Diagnostics Laboratory of IRCCS IRST during the period 2007–2019. Such methodologies included: Sanger sequencing, MassARRAY Sequenom, real-time PCR, pyrosequencing, and panel NGS assays.

### 2.3. Statistical Analysis

Data were summarized by using median and minimum and maximum values or interquartile range (IQR), as appropriate for continuous variables and by means of natural frequencies and percentages for categorical ones.

TMB was classified in two ways: one using the threshold of 10 mut/Mb for discriminating low (<10 mut/Mb) and high (≥10 mut/Mb) values [16]; the other using the thresholds of 6 and 20 mut/Mb to differentiate between low (≤5 mut/Mb), intermediate (6–19 mut/Mb), and high (≥20 mut/Mb) values [17].

The association between the number of pathogenic alterations or TMB (as a continuous variable) and the demographic and clinical covariates was assessed using the Mann–Whitney U test or the Kruskal–Wallis test, as appropriate. The association between TMB (as a categorical variable) and the demographic and clinical covariates was assessed using the Chi-square test or the Fisher exact test, when appropriate.

Concordance between the results from Foundation One CDx and from routinely used assays for the detection of gene variants, copy number alterations, and rearrangements were assessed by means of the proportion of observed agreement, p_0_, the Cohen’s kappa, the PABAK index [18], and agreement charts [19].

For all the analyses, a two-sided *p*-value < 0.05 was considered statistically significant. Analyses were carried out with R version 3.6.2 statistical software [20] (http://www.r-project.org/index.html, accessed on 20 February 2022); the oncoprint was obtained using the ComplexHeatmap package and the agreement charts from the vcdExtra package.

## 3. Results

### 3.1. Case Series Characteristics

Of the identified 368 patients, 225 (61%) were excluded as no residual histologic material was available from either the Pathology Units of “Morgagni-Pierantoni” Hospital of Forlì and “M. Bufalini” Hospital of Cesena, which had already been used for routinely molecular diagnostics evaluations, and 31 (8%) were excluded as only cytological samples were available. In total, 112 cases (30%) were sent for molecular analysis, of which 93 (83%) resulted in a valuable result. For 64 patients, a routinary molecular diagnostic analysis was performed (Figure 1). With regard to molecular analysis, the MASSArray Sequenom methodology was the most frequently used (>90% of cases analyzed with this approach), whereas, for translocation analysis, FISH and IHC were the methodologies usually used. Clinicopathological characteristics of the whole case series are described in Table 1. In particular, all patients were ≤65 years old, with a median age of 59.6 (IQR: 8.4). Seventy-seven (82.8%) patients had adenocarcinoma (ADC), thirteen (13.9%) had a squamous cell carcinoma (SCC), and three patients had undifferentiated carcinoma. Forty-eight (51.6%) were females, and forty-eight (48.4%) were males. Seventy patients (75.3%) had a smoking history, with fifty (59.5%) current and twenty (23.8%) former smokers, whereas fourteen (16.7%) patients were never smokers. At diagnosis, 37 patients (40.2%) had a resectable stage IA-IIIA tumor, whereas 55 (59.8%) had an advanced stage tumor.

### 3.2. Sequencing Results

One hundred and twelve patients were molecularly characterized by Foundation One CDx assay. Of these, 19 failed the molecular analysis due to inadequate quality and/or quantity of extracted DNA, resulting in 93 patients with a valuable and interpretable result. A total of 1200 short variants were identified, of which 271 were called “as known” or “likely” somatic by Foundation One CDx assay. Moreover, 273 CNAs and 23 rearrangements were identified. Globally, in all cases, at least one pathogenic alteration (mutation, CNA, or rearrangement) was identified, with 42% of cases carrying more than five known somatic events (Figure 2A). In Figure 2B–D are reported the number and frequency of the different types of alterations. The most frequent altered genes were TP53 (64.55%), followed by KRAS (44.1%), STK11 (26.9%), CDKN2A (21.5%), CDKN2B (14.0%), EGFR (16.1%), and RB1 (10.8%) (Figure 3). No patients showed microsatellite instability. The median TMB value was 7.57 mut/Mb (range 0–55.5), and it was not evaluable for three patients. Thirty-two patients (34.4%) had a low TMB, fifty-three (56.9%) had an intermediate TMB, and five (5.4%) had a high TMB status. Considering the cut-off of 10 mut/Mb, 62 (68.9%) patients showed a TMB < 10 mut/Mb, whereas 28 (31.1%) showed a TMB ≥ 10 mut/Mb.

The presence of specific mutations resulted as significantly associated with TMB status. Notably, STK11 and KRAS mutations were significantly associated with a higher TMB status by considering it as a continuous variable (*p* = 0.019, *p* = 0.004, respectively). Conversely, EGFR and EML4-ALK alterations were more frequently found in tumors with low TMB (*p* = 0.019 and *p* < 0.001, respectively).

### 3.3. Frequency of the Principal Targetable Alterations

Considering the principal targetable alterations in NSCLC, in accordance with OncoKB classification [21] and considering those alterations with an approved or under investigation targeted drug available, a potentially targetable gene alteration was detected in 67 (72%) of patients. Regarding gene variants, 11 (11.8%) patients showed an EGFR mutation, of which seven had an exon 19 deletion (three with a concomitant T790M mutation), two patients showed an L858R mutation, one an exon 19 insertion, and one patient showed a G719A mutation. Thirty-six (38.7%) patients showed a KRAS mutation, of which 15 (41.7%) had the targetable G12C mutation. Five (5.4%) patients had a BRAF mutation, all localized at exon 11. Four (4.3%) patients showed a HER2 mutation, two exon 20 insertions, one exon 20-point mutation (L755P), and one exon 19-point mutation (S310F). Six (6.5%) patients showed a PIK3CA mutation, three patients had an exon 9-point mutations (E542K), two patients had an exon 20-point mutation (one H1047R and one had two concomitant point mutations, L1067F and M1004I), and one had an exon 1 mutation (K111E). Finally, one patient showed an exon 14 MET mutation (3028 + 1G > A) (Appendix A). Among rearrangements, seven known fusion genes were identified: six were EML4-ALK, while one was a RET-KIF5B fusion. One patient carrying an EML4-ALK translocation also showed an ALK point mutation at exon 22 (E1161K). Regarding gene CNA, two patients showed MET amplification. Moreover, 18 (19%) of patients showed concomitant targetable alterations. Notably, all four patients showing a BRAF mutation had a concomitant targetable alteration (two patients had a concomitant KRAS point mutations, and two had a PIK3CA point mutation), and all five patients with a PIK3CA mutation had a concomitant alteration (two had concomitant BRAF mutations, one had an EGFR mutation, one had an ERBB2 mutation, and one had both an EGFR and an ERBB2 mutations).

Figure 4 shows the frequency of targetable alterations ordered by increasing TMB values. A higher KRAS mutation frequency seemed to be present for patients with increasing values of TMB, whereas EGFR mutations and ALK alterations were more frequently observed for patients with lower values of TMB. As expected, never smokers were more frequent in low TMB patients.

### 3.4. Molecular Pathogenic Alterations and Clinicopathological Characteristics of Patients

A significant association was found between the number of pathogenic variants and smoking habit. Current smokers, particularly, had a median number of variants/patients of 3 (range 1–6), former smokers had a median of 2.5 (range 1–9) alterations, whereas a median number of 1 (range 0–6) alteration was found in never smokers (*p* = 0.018). Similarly, the smoking habit was significantly associated with the number of pathogenic rearrangements. In particular, in current smokers, 60.8% of patients did not show any rearrangement, whereas this percentage was lower in former smokers (27.0%) and in never smokers (9.0%) (*p* = 0.006). A statistically significant association was also observed between the presence of rearrangements and age at diagnosis. Notably, a higher percentage of rearrangements was observed in younger patients (defined as patients with ≤50 years) with respect to older ones (*p* = 0.033). No statistically significant differences were observed between pathogenic CNA and any clinicopathological characteristics of patients.

Regarding TMB, a significantly higher median TMB was observed in former smokers (median 10.1 mut/Mb, range 3.78–55.48), followed by smokers (median 7.57 mut/Mb, range 0–30.26) and never smokers (median 2.52 mut/Mb, range 0–10.09), *p* < 0.001. By categorizing TMB as low, intermediate, and high, we found that 60% of patients with high TMB were current smokers, and 40% were former smokers, whereas none of the never smokers had a TMB high (*p* = 0.004).

### 3.5. Comparison between Routinary Molecular Analysis and Foundation One CDx Assay

Of the 93 cases analyzed by Foundation One CDx assay, 68 were previously characterized by routinary molecular diagnostics analysis. Considering the range of years considered (2004–2019), different markers were analyzed using different methodologies. In particular: 62 patients were characterized for EGFR status, of which 45 by MassARRAY/Sequenom, 16 by pyrosequencing, and 1 patient by NGS using the Myriapod® NGS 56G Onco panel (Diatech Pharmacogenetics); 35 patients were characterized for KRAS status (34 by Massarray/Sequenom and 1 by NGS); 26 patients were characterized for BRAF (25 by MassArray/Sequenom and 1 by NGS); 23 patients were characterized for HER2 (22 by MassArray/Sequenom and 1 by NGS); 44 patients were characterized for ALK (24 by immunohistochemistry, 10 by fluorescence in situ hybridization (FISH), 9 by FISH plus IHC, and 1 by NGS); 29 patients were characterized for ROS1 (23 by FISH, 5 by IHC, and 1 by NGS); 12 patients were characterized for MET (10 by FISH, 1 by NGS, and 1 by Sanger sequencing).

The concordance between Foundation One CDx analysis and the methodologies used for routinary molecular characterization at IRST laboratory was assessed.

Appendix A shows the agreement chart for the main alterations, that is, EGFR, KRAS, and HER2 mutations, and EML4-ALK translocations. Perfect agreement was observed regarding EGFR and HER2 mutations (p_0_ = 1.0, kappa = 1.0 and PABAK = 1.0 for both genes). These statistics were obtained for 60 and 23 patients, respectively. That is, only for those patients for whom a diagnostic determination was asked by the clinician.

Substantial agreement was observed for KRAS and ALK alterations (p_0_ = 0.9, kappa = 0.7 and PABAK = 0.7 for KRAS—35 patients—and p_0_ = 0.9, kappa = 0.6 and PABAK = 0.9 for ALK—44 patients). With respect to other alterations such as those in BRAF, ROS1, and MET, 25, 28, and 12 samples were analyzed with both methods. In all cases, either Foundation One CDx and routinary laboratory diagnostic assays classified all patients as wild type; results are not shown.

## 4. Discussion

All patients with advanced NSCLC require a fast-track molecular determination of biomarkers to decide on the first-line targeted therapy, immunotherapy, or immunotherapy/chemotherapy combination. Molecular features of lung cancer are also influenced by environmental, familial, and lifestyle factors, and environmental factors often display nonlinear relationships with cancer outcomes [22,23]. In this regard, it has been recently highlighted that environmental factors such as age and tobacco smoking are strictly related to NSCLC molecular features [23]. Moreover, gene–gene and gene–environment relations have been proved to influence response and toxicity to systemic therapies [24]. Over the last years, we have witnessed an evolution with regard to the approaches to be used for molecular characterization, passing from single gene methodologies such as Sanger sequencing, pyrosequencing, and real-time PCR to multigene panel methods such as MassARRAY and NGS [12]. Nowadays, with increasing evidence of targetable molecular alterations, multigene panels are the preferential approach to be applied in molecular diagnostics clinical practice. In our study, we compared results obtained by a case series of young NSCLC, characterized using different routinary methodologies, and characterized by using a large gene panel assay. The most frequent methodology used in routine molecular diagnostics was MASSArray Sequenom, characterized by a range of sensitivity quite similar to that obtained by NGS methodologies [25]. In agreement with this, a perfect concordance was observed for EGFR mutations. Moreover, our results were in perfect agreement also in relation to EML4-ALK translocation, for which the NGS approach gave the same results obtained using in situ methodologies (FISH/IHC). However, the multiple gene assay allowed for simultaneous information, which could, today, be very important in clinical practice. By using the multigene test assay, a potentially targetable mutation was detected in about 70% of patients.

The Food and Drug Administration (FDA) has already approved many drugs targeting other gene alterations than common EGFR activating mutations and ALK translocations. These include ROS1 rearrangements, BRAF V600E, NTRK rearrangements, MET exon 14 skipping mutations, and RET rearrangements. However, tyrosine kinase inhibitors, monoclonal antibodies, and conjugates are also under development for further gene targets, such as uncommon EGFR mutations, HER2 mutations, KRAS G12C mutation, and NRG1 rearrangements [26].

Regarding KRAS mutation, we observed a higher frequency of mutations by using the multiple gene NGS assay. This could be due to the high sensitivity of the methodology and to the fact that by routinary standard approaches, the KRAS test was performed only on a limited number of cases. Recent studies have highlighted the high efficacy of novel therapeutic drugs directed against the KRAS G12C mutation [27,28]. In our case series, 16% of patients had this mutation, with the possibility of being included in ongoing clinical trials and having more therapeutic chances. Moreover, of the 15 patients carrying a KRAS G12C mutation, six (40%) showed a concomitant STK11 mutation. Recent evidence showed that the concomitance of G12C and STK11 mutations increase the sensitivity to anti-KRAS G12C therapies, such as sotorasib or adagrasib [29], and decreases the sensitivity to immunotherapeutic agents [30], confirming the importance of a multigenic tumor characterization before any treatment decision making. A high TMB status was identified in about 30% of cases, considering the cut-off of 10 mut/Mb [16]. However, by dividing TMB levels into three groups based on the Foundation Medicine official reports, only 5% of patients were classified with a high TMB ≥ 20 mut/Mb. In accordance with previous literature results, high TMB tumors were more frequently observed in former or current smokers [31]. Moreover, STK11 and KRAS mutations were significantly associated with high TMB tumors, whereas EGFR and ALK alterations were frequently associated with a low TMB status.

In Europe, national clinical guidelines for molecular testing and targeted therapy refer to ESMO/NCCN guidelines and are generally tailored to national healthcare models and resources. Limitations of the use of NGS may include different aspects such as turnaround time, lack of reimbursement of the test, restrictions on the possibility of having the targeted drugs for the identified targetable alterations available, the difficulty in results interpretation. Once an NGS report has been produced and a targetable alteration has been found, some therapeutic chances could be open for the patient. ESMO has developed a system that defines the levels of evidence supporting the use of a targeted drug matching with the molecular alteration found. Through the tool known as the ESMO Scale of Clinical Actionability for molecular Targets (ESCAT), the appropriateness of specific alteration-based treatment usable in clinical practice could be defined [10], helping oncologists in their clinical decision making. The creation of the Molecular Tumor Board (MTB) within the different institutions, involving clinicians, molecular pathologists, molecular biologists, bioinformaticians, and geneticists, could also facilitate the usage and interpretation of NGS reports. All these aspects, together with recent evidence demonstrating that NGS approaches are less costly with respect to single-gene testing approaches [32] by considering different scenarios of clinical practice evolution, are leading more and more towards an ever-wider use of multitarget technologies.

## 5. Conclusions

From our study emerged the importance of a multipaneled NGS approach, underlying how different information can simultaneously be available and useful for treatment decision making. By comparing a multigenic approach with standard mono- or limited-gene ones, it was evident that the first one was able to reveal molecular information, which is emerging today as very important in clinical practice. Considering that precision medicine is an effective approach for lung cancer treatment, the application of wide molecular approaches could not only help clinicians to better tailor patients’ clinical management but also achieve information about genes that could influence patients’ outcomes. Although different issues must be addressed, the multipaneled NGS approaches represent the future for the routine molecular characterization of NSCLC patients.

## Figures and Tables

**Figure 1 cancers-14-02352-f001:**
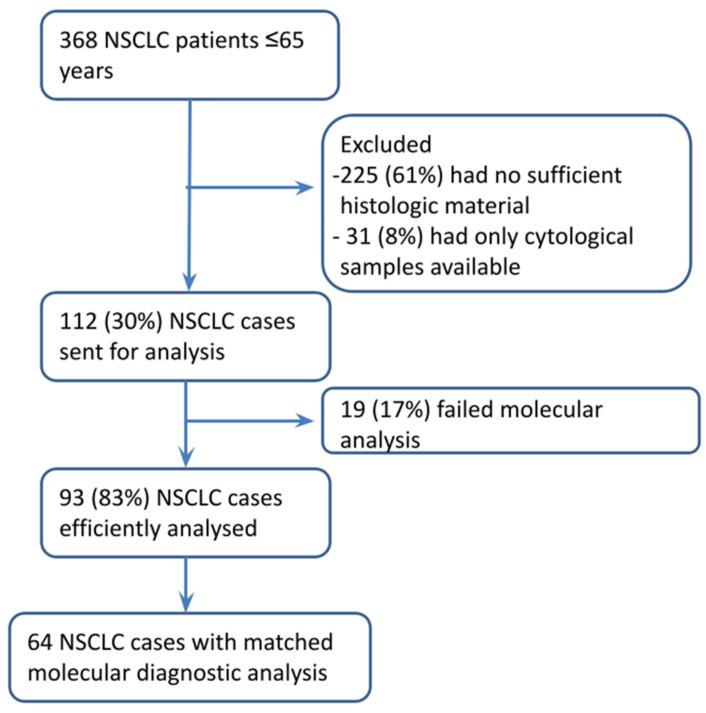
Flow chart of patient enrolment.

**Figure 2 cancers-14-02352-f002:**
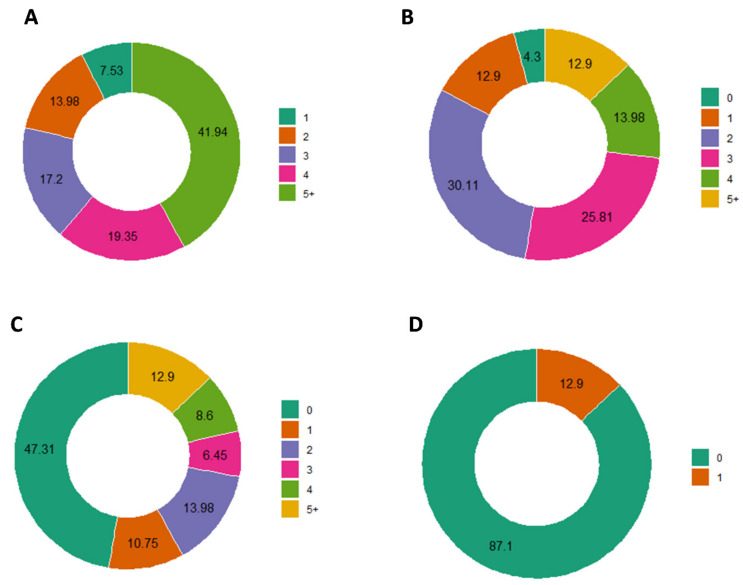
Frequency of patients with: (**A**) one or more concomitant alterations, considering all types of gene alterations, (**B**) one or more gene variants, (**C**) one or more CNA, and (**D**) one or more rearrangements.

**Figure 3 cancers-14-02352-f003:**
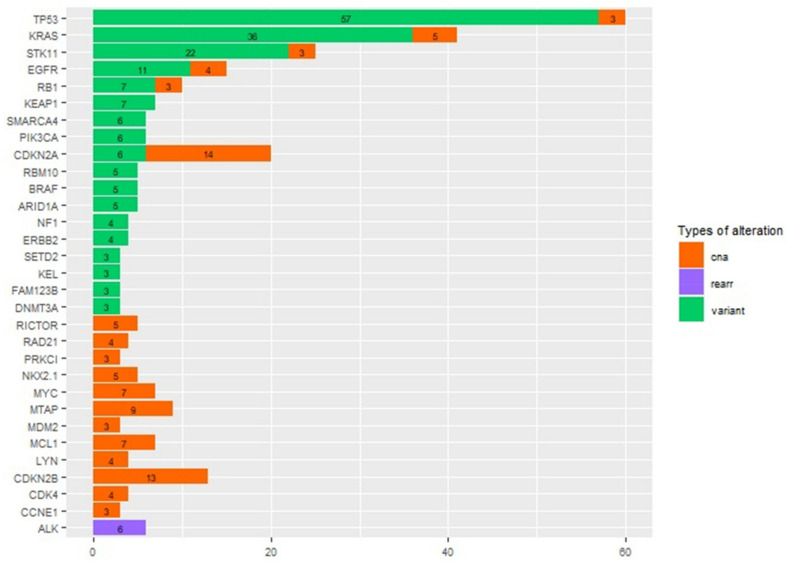
Number of patients with alterations in different genes, with specified the type of alteration for each gene.

**Figure 4 cancers-14-02352-f004:**
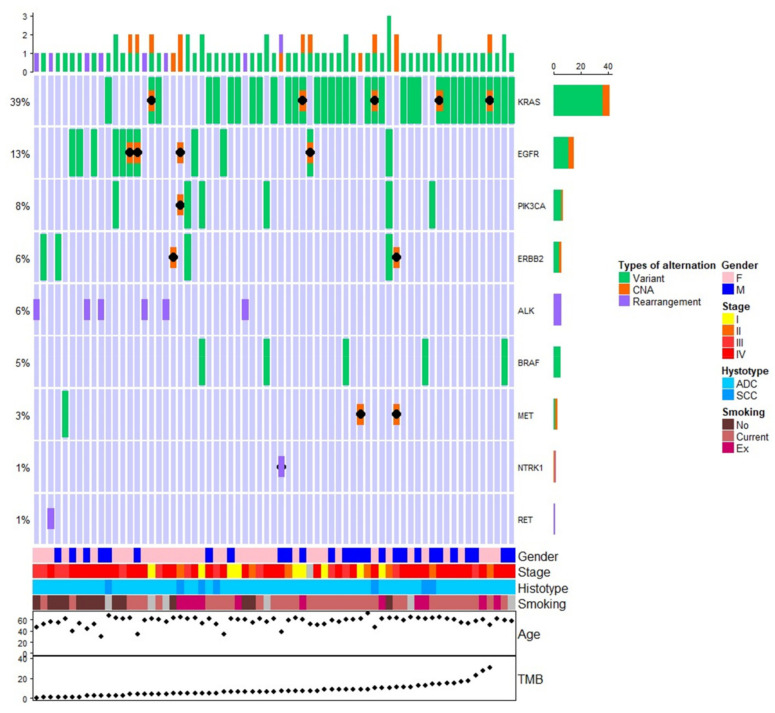
Oncoprint plot showing the distribution of targetable alterations and clinic-pathological patient’s characteristics ordered for TMB value (from lowest to highest value).

**Table 1 cancers-14-02352-t001:** Clinico-pathological patient’s characteristics (*n* = 93).

	*n*	%
Gender		
F	48	51.6
M	45	48.4
Age at diagnosis (yrs)		
Median [IQR]	59.64 [8.4]	
Smoking habit *		
Non-smoker	14	16.7
Current smoker	50	59.5
Ex-smoker	20	23.8
Histotype		
Adenocarcinoma	77	82.8
Squamous carcinoma	13	14.0
Other	3	3.2
Stage at diagnosis *		
IA	2	2.2
IB	10	10.9
IIA	4	4.3
IIB	9	9.8
IIIA	12	13.0
IIIB	7	7.6
IV	48	52.2
Cardiovascular comorbidities		
Hypertension	26	29
Ischemia/coronary artery disease	7	7.7
Stroke/thrombosis	8	8.9
Atrial fibrillation	1	1.1
Other	2	2.2
2 or more	8	8.8
NA	4	
Pulmonology comorbidities		
COPD	26	29.9
IPF	1	1.1
Asthma	2	2.3
Other	1	1.1
NA	7	
Metabolic comorbidities		
Obesity	1	1.1
Diabetes 1/2	8	9.0
Metabolic syndrome	10	11.4
2 or more	1	1.1
NA	6	
Viral infections		
HBV	1	1.1
HCV	1	1.1
NA	7	
2 or more comorbidities	30	31.9

IQR—interquartile range. * Sum does not add up to the total due to missing values. COPD—chronic obstructive pulmonary disease; IPF—idiopathic pulmonary fibrosis; NA—not available.

## Data Availability

The datasets generated and/or analyzed during the current study are available from the corresponding author on reasonable request.

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
