# Peer review of "Wide Next-Generation Sequencing Characterization of Young Adults Non-Small-Cell Lung Cancer Patients"

_cancers, 2022, doi:10.3390/cancers14102352_

Round 1
Reviewer 1 Report
The manuscript presented by Ulivi is interesting and contributes to the generation of new knowledge on the subject. The introduction is consistent with the purpose of the study. The methodology is sufficient and updated, and the results support the discussion. However, I have the following comments.
I. Major Comments:
1. In the introduction, I suggest briefly including a paragraph regarding the role of environmental factors, lifestyles, and aging in the risk of developing lung cancer.
2. Methodology: Include inclusion and exclusion criteria. Did the recruited patients have a history of other pathologies (obesity, DM2, high blood pressure, etc.)?
3. Results. Table 1. Include age, weight, height and BMI of the patients. I also suggest including more clinical information on the patients. This information is important to have a better understanding of the sample (characterization of the population).
4. The discussion is good, but I suggest: - Briefly discuss possible genetic or environmental interactions, and their relationship to lung cancer. - Include a figure that allows a better understanding of the results
5. Considering the results, improve the clinical projections of the study.
II. Minor comment:
1. Improve the writing of the objective of the study.
Author Response
Reviewer 1
The manuscript presented by Ulivi is interesting and contributes to the generation of new knowledge on the subject. The introduction is consistent with the purpose of the study. The methodology is sufficient and updated, and the results support the discussion. However, I have the following comments.
- Major Comments:
1. In the introduction, I suggest briefly including a paragraph regarding the role of environmental factors, lifestyles, and aging in the risk of developing lung cancer.
Re: We thank the Reviewer for the comment; in the introduction, we briefly described the main environmental and lifestyle factors associated to lung cancer (lines 57-59).
Methodology: Include inclusion and exclusion criteria. Did the recruited patients have a history of other pathologies (obesity, DM2, high blood pressure, etc.)?
Re: We thank the Reviewer for the comment. We included main inclusion criteria for the study (line 129-131). As suggested, cardiovascular, metabolic, lung and viral pathologies at diagnosis were collected and reported (Table 1).
Results. Table 1. Include age, weight, height and BMI of the patients. I also suggest including more clinical information on the patients. This information is important to have a better understanding of the sample (characterization of the population).
Re: We thank the Reviewer for the comment. Unfortunately, it was not possible to retrieve all medical information required, for missing for most of patients in the medical records. We highlight that age of patients was provided, as an inclusion criteria. On the other hand, we retrieved information about different comorbidities incidence for the study population, to better provide patients characterization (see Table 1).
The discussion is good, but I suggest: - Briefly discuss possible genetic or environmental interactions, and their relationship to lung cancer. - Include a figure that allows a better understanding of the results.
Re: We thank the Reviewer for the comment. In the Discussion paragraph, we added the required information (lines 354-360). For what concerns the required figure, 5 figures (4 along the text and 1 in the Supplementary material) are yet included in the manuscript, to present study results. We kindly ask the Reviewer which type of figure could we add to improve results presentation.
Considering the results, improve the clinical projections of the study.
Re: We thank the Reviewer for the comment. We added a sentence in the conclusions paragraph to better address the clinical projections of the study (lines 444-448).
- Minor comment:
1. Improve the writing of the objective of the study.
Re: We thank the Reviewer for the comment. We reorganized the sentence describing the aim of the study in the Introduction paragraph (line 118).
Reviewer 2 Report
Have NGS data resulting from this study been deposited in full (i.e. SRA/GEO)? This will truly make the study a major asset and an important resource for the scientific community.
Minor comments:
- lines 55-56: This is the number of new cases and deaths since when/by when/ within what time span? Please, clarify.
- Number of patients is at times indicated in numbers at times in letters - please be consistent throughout the text.
- lines 177-184: To which patient group these %s refer to? Please, clearly indicate this in the main text.
Author Response
Reviewer 2
Have NGS data resulting from this study been deposited in full (i.e. SRA/GEO)? This will truly make the study a major asset and an important resource for the scientific community.
Re: We thank the Reviewer for the useful comment. We agree with the Reviewer that this will improve our manuscript; we asked Roche to upload raw data on one of the two platforms, and we are awaiting that to be performed. In order to not postpone too much our submission, we resubmitted the manuscript, but the upload of raw data is warranted.
Minor comments:
lines 55-56: This is the number of new cases and deaths since when/by when/ within what time span? Please, clarify.
Re: We thank the Reviewer for the comment. We accordingly addressed the information (lines 55-56 ).
Number of patients is at times indicated in numbers at times in letters - please be consistent throughout the text.
Re: We thank the Reviewer for the comment. We addressed this issue along the text (lines 201-208).
lines 177-184: To which patient group these %s refer to? Please, clearly indicate this in the main text.
Re: We thank the Reviewer for the comment. These %s refer to the whole case series, as describing Table 1. We highlighted that along the text (line 201).
Round 2
Reviewer 1 Report
Authors answered all my comments. Therefore, the manuscript can be accepted in the present form.